# The First Lactate Threshold Is a Limit for Heavy Occupational Work

**DOI:** 10.3390/jfmk5030066

**Published:** 2020-08-25

**Authors:** Patrick Fasching, Stefan Rinnerhofer, Georg Wultsch, Philipp Birnbaumer, Peter Hofmann

**Affiliations:** 1Private Clinic Maria Hilf, Radetzkystraße 35, 9020 Klagenfurt, Austria; fasching.spowi@gmail.com; 2Center for Occupational Medicine AMEZ, Hergottwiesgasse 149, 8055 Graz, Austria; stefan.rinnerhofer@gmail.com (S.R.); georg.wultsch@medunigraz.at (G.W.); 3Medical University of Graz, Auenbruggerplatz 2, 8036 Graz, Austria; 4Institute of Human Movement Science, Sport and Health, University of Graz, Max-Mell-Allee 11, 8010 Graz, Austria; philipp.birnbaumer@uni-graz.at

**Keywords:** heavy work, intensity threshold, individualization, maximum oxygen uptake, work strain

## Abstract

Long-term heavy physical work often leads to early retirement and disability pension due to chronic overload, with a need to define upper limits. The aim of this study was to evaluate the value of the first lactate threshold (LTP_1_) as a physiological marker for heavy occupational work. A total of 188 male and 52 female workers performed an incremental cycle ergometer test to determine maximal exercise performance and the first and second lactate (LTP_1_; LTP_2_) and ventilatory thresholds (VT_1_; VT_2_). Heart rate (HR) recordings were obtained during one eight-hour shift (HR_8h_) and oxygen uptake was measured during 20 minutes of a representative work phase. Energy expenditure (EE) was calculated from gas-exchange measures. Maximal power output (P_max_), maximal oxygen consumption (VO_2 max_) and power output at LTP_1_ and LTP_2_ were significantly different between male and female workers. HR_8h_ was not significantly different between male and female workers. A significant relationship was found between P_max_ and power output at LTP_1_. HR_8h_ as a percentage of maximum HR significantly declined with increasing performance (P_max_:r = −0.56; *p* < 0.01; P_LTP1_:r = −0.49; *p* < 0.01). Despite different cardio-respiratory fitness-levels; 95.4% of all workers performed their usual work below LTP_1_. It is therefore suggested that LTP_1_ represents the upper limit for sustained heavy occupational work; which supports its use to determine work capability and assessing the limits of heavy occupational work.

## 1. Introduction

Chronic overload during long-term heavy occupational work was shown to be related to early retirement due to cardiovascular diseases, disability pension and all-cause mortality [1,2,3,4]. It is therefore of great interest to determine the limits for heavy work and the fitness requirements of workers performing such heavy work [5]. Additionally, the individual fitness status of workers has been prescribed to have a significant impact on health. It was shown that with each 10% increase in individual strain during work, the risk of heart attacks in healthy male individuals increases by 18% indicating the importance of individual performance [6] and normal weight [1]. In summary there is strong evidence from the literature, that a permanent high strain level due to heavy work is harmful to health. To prevent workers from overload, physiological limits for heavy work have been set up. These limits define the maximal sustainable workload, which should not be exceeded permanently during the working time and are usually defined as upper limits for physiological performance [7,8,9].

The most frequently applied variables as a limit for heavy work are heart rate (HR), oxygen uptake (VO_2_) and work-related energy expenditure (WEE). Most limits, however, only prescribe the external workload, but do not give an adequate measure of the individual strain of a person. An accepted limit for heavy work was prescribed to be 110 b·min^−1^ [10]. Working below this limit should prevent workers from developing cumulative fatigue. However, several studies showed that work can also be sustained above this limit [11,12]. Additional frequently applied limits are prescribed as a percentage of maximal oxygen uptake (% VO_2 max_) or maximal heart rate (% HR_max_) which represent the individual strain of a person. An accepted% VO_2 max_ limit for heavy work was prescribed to be 33% [7]. However, this value does not represent an individual limit which has already been discussed [13]. Additionally, it was shown, that the same % VO_2 max_ [14] or % HR_max_ [15], do not represent the same strain when related to validated threshold markers [16]. For instance, an exercise intensity of 85% HR_max_ causes different metabolic and cardio-respiratory responses across individuals [17], indicating that % VO_2 max_ and % HR_max_ targets are also limited [5].

It is therefore essential to define a marker, which is a valid parameter of individual strain, physiologically justifiable and can be applied to heavy work. A suitable physiological marker for high-volume continuous loads is the first increase in blood lactate concentration (LTP_1_) in an incremental cycle ergometer test originally prescribed as “anaerobic threshold” [18]. Later on, a three-phase energy supply according to the shuttle theory of Brooks [19], has been prescribed including a second threshold of lactate (LTP_2_) or ventilation (VT_2_) [16,17]. Figure 1 shows the time course of blood lactate concentration [20,21] and both thresholds (LTP_1_, LTP_2_) dividing three distinct phases of energy supply which have been validated recently [22].

Due to the long duration of occupational work, usually eight hours or even longer, the low intensity range is most relevant. To sustain long-term work low cardio-respiratory and metabolic strain are essential conditions to avoid acute and chronic overload. Intensities below LTP_1_ represent a balance of lactate production and oxidation with low cardio-respiratory strain. Exceeding LTP_1_, leads to a significant increase in most if not all physiological variables indicating a stress reaction clearly limiting duration [23]. Below LTP_1_ there is just a withdrawal of the parasympathetic influence reaching its minimum at the LTP_1_ [24]. Above LTP_1_, sympathetic drive increases, indicated by a rise in catecholamines and related variables [17,22]. Moser et al. [22], nicely showed that catecholamine levels did not increase above baseline values if workload was below LTP_1_ during constant load cycling and just slightly increased with high intensity intervals if the mean load was below LTP_1_. It is therefore obvious that exceeding LTP_1_ clearly reduces time to fatigue [23]. Furthermore, workloads above LTP_1_ rely on a greater carbohydrate consumption, also limiting time to fatigue, as an energy deficit will be reached within a few hours [25]. Working at intensities below LTP_1_, prolongs performance time substantially which has been shown in ultra-distance athletes cycling 24 h at a workload clearly below LTP_1_ [26]. Similar has been shown by Tipton et al. [9], prescribing the physical performance limit with no increase in blood lactate corresponding to the LTP_1_. Additionally, Kunutsor et al. [27], showed that a higher oxygen-uptake capacity at a threshold equivalent to LTP_1_ was associated with a lower risk of cardiovascular disease and all-cause mortality. However, the absolute limits for specific occupations still need to be analyzed in detail. We therefore suggest, that LTP_1_ may serve as an individual upper limit for heavy occupational work. We hypothesized that all workers stay below the LTP_1_ limit during their work shift and that the relative strain is related to individual performance. Therefore, the aim of this study was to evaluate LTP_1_ as a physiological marker for the upper limit of heavy occupational work in male (m) and female (f) workers.

## 2. Materials and Methods

The study was approved by the local ethics committee of the Medical University (EK number 26-488 ex 13/14, 10.09.2014). Informed consent was obtained from all subjects who underwent a comprehensive medical examination by the leading physician of the study before the start of the tests. In addition to a detailed medical history and a survey on exclusion criteria such as medication, pre-existing illnesses and movement restrictions, a resting electrocardiogram (ECG) was performed in all participating subjects. A total of 240 male (m: *N* = 188) and female (f: *N* = 52) subjects from six different occupations (metalworkers, workers in butchery, garbage collectors, cabinet maker, physiotherapists and masseurs) classified as “heavy work” by law, participated in the study (Table 1).

### 2.1. Incremental Cycle Ergometer Test

Subjects performance was determined by means of a maximal incremental cycle ergometer exercise test using an electronically braked, computer-controlled ergometer (Monark 839E, Monark Exercise AB, Vansbro, Sweden). Workload started at 40 Watt (W) (m) or 20 W (f) and was increased by 20 W (m) or 15 W (f) every minute. This protocol is the recommended protocol of the local Society of Cardiology to determine maximal power output (P_max_) among young and healthy people [28]. The heart rate (HR) and a 12-lead ECG were measured continuously, and blood pressure was manually measured at each workload for safety reasons (not shown). At rest, after each workload and after 3 minutes of recovery capillary blood (20 μL) was taken to determine blood lactate concentration (BIOSEN S-Line, Lab +, EKF Diagnostic GmbH, Barleben, Germany) for further analyses. Both, LTP_1_ and LTP_2_ were assessed by means of computer-aided linear regression breakpoint analysis. LTP_1_ was defined as the first increase in blood lactate concentration (La) above baseline [29], according to the “anaerobic threshold” [18]. Respiratory gas exchange was measured continuously throughout the complete test (Metalyzer 3B-R2, Cortex Biophysik, Germany) and the first (VT_1_) and the second (VT_2_) ventilatory turn points [30], were determined with a standard software (MetaSoft version 3.9.7 SR5, Cortex Biophysik, Leipzig, Germany). VT_1_ was defined as the first increase in ventilation accompanied by an increase in the equivalent for oxygen uptake (VE/VO_2_) without an increase in the equivalent for carbon dioxide output (VE/VCO_2_). VT_2_ was defined as the second abrupt increase in ventilation accompanied by an increase in both VE/VO_2_ and VE/VCO_2_. P_max_ and maximal oxygen uptake (VO_2 max_) were compared to standard values (P_max-target_, VO_2 max-target_) by age, height, weight and gender [28].

### 2.2. Measurements during Work

HR was recorded continuously during the eight hours of working time (HR_8h_) with a commercial heart rate monitor (Polar S610, Polar Electro, Kempele, Finland). Oxygen uptake (VO_2_) and carbon dioxide output (VCO_2_) were measured during 20 min of a representative work phase (VO_2 20_, VCO_2 20_) with a portable device (Metamax 3B, Cortex Biophysik, Leipzig, Germany). The calibration of the system was carried out before each test according to the specifications of the manufacturer. Pressure calibration for ambient air pressure was carried out with a digital reference barometer and the volume sensor (Triple-VR turbine) was calibrated with a 3-L calibration pump (Hans Rudolph, Inc., Shawnee, Kansas, USA). Gas-sensor calibration was carried out with a mixed test gas (15 vol% O_2_, 5 vol% CO_2_, balance N_2_) and ambient air according to the calibration manual of the manufacturer (Cortex, MetaMaxR 3B and MetalyzerR 3B). Working energy expenditure (WEE) was calculated by means of indirect calorimetry using the 20-min gas-exchange measurement during a representative work phase [31] and adjusted for mean 8 h HR.

### 2.3. Statistical Analysis

Data were processed with Microsoft Excel 2013 and Winstat Statistics Version 3.1 (Winstat, Kalmia Corp., Cambridge, MA, USA). The graphic representations were edited with GraphPad Prism software (GraphPad version 5.01, San Diego, CA, USA). All data are presented as mean values with standard deviation (±SD) and data were evaluated for normal distribution (Kolmogorov–Smirnov test). Relationships between variables were evaluated by linear regression analyses and Pearson’s correlation coefficient was determined where appropriate. Differences between groups were analyzed by means of variance analysis (ANOVA) and Tukey’s post hoc analysis and Pearson’s product moment correlation were used to calculate the relationship between variables. The effect size was calculated as Hedges’ g [32]. An error probability less than 5% (*p* < 0.05) was accepted.

## 3. Results

Male and female workers were not significantly different for age and body mass index (BMI), but for height and weight (Table 1). Male subjects were slightly overweight compared to normal weight female subjects [33].

### 3.1. Incremental Cycle Ergometer Test

Results of the incremental cycle ergometer tests are presented in Table 2 and Table 3. Male subjects had a significantly higher P_max_ and VO_2 max_ but were significantly lower compared to standard target values (Table 2). No significant differences were found for HR_max_ and the maximal respiratory exchange ratio (RER_max_), suggesting a similar strain for both male and female subjects. Maximal blood lactate concentration (La_max_) was slightly, but significantly lower in female subjects. Male subjects reached significantly higher power output and oxygen-uptake values at LTP_1_ (P_LTP1_, VO_2 LTP1_) and VT_1_ (P_VT1_ VO_2 VT1_), but lactate and ventilatory thresholds were not significantly different. P_LTP1_ as a percentage of P_max_ was significantly higher in male, but VO_2 LTP1_ as a percentage of VO_2 max_ was significantly higher in female subjects. (Table 3). HR values at LTP_1_ were significantly higher in female subjects. Additionally, % HR_max_ reached significantly higher values for LTP_1_ and VT_1_ in the female group. No significant differences between male and female subjects were found for blood lactate concentration at LTP_1_ and VT_1_. Power output and oxygen uptake at LTP_2_ and VT_2_ were significantly higher for male compared to female subjects, but the respective percentages of VO_2 max_ were significantly lower in the male group. No significant differences between groups were found for HR and La at LTP_2_ and VT_2_.

### 3.2. Measurements during Work

The HR_8h_ of both female and male subjects were below the LTP_1_ (Table 4). The time course of HR for a highly fit (LTP_1_ = 125 W) and a low fit (LTP_1_ = 81 W) subject, during eight hours of working showed, that the high-fit and the low-fit person achieved a mean HR (109 vs. 123 b·min^−1^) which corresponded to 100% and 101% of the individual HR at the LTP_1_, respectively. Despite different fitness, both subjects achieved almost identical relative strain values with significantly different absolute HR values (*p* < 0.05). VO_2 LTP1_ was significantly related to VO_2 max_ (r = 0.86, *p* < 0.01) similar to P_LTP1_ and P_max_ (r = 0.87, *p* < 0.01) (Figure 2). HR_8h_ as a percentage of HR_max_ was significantly related to P_max_ (r = −0.56, *p* < 0.01), VO_2 max_ (r = −0.41, *p* < 0.01) and P_LTP1_ (r = −0.49, *p* < 0.01). No statistically significant relationship was found between age and P_LTP1_ (r = 0.02, *p* = 0.37) and VO_2 20_ (r = −0.04, *p* = 0.27).

Figure 3 shows the relationship between HR_8h_ as a percentage of HR_max_ and power output as a percentage of target standard power output (P_max-target_) from the incremental cycle ergometer test as well as the mean individual HR limit at LTP_1_ (100% HR_LTP1_) during the working activity. In total, 95.4% of all female and male workers stayed below the individual LTP_1_ during their working hours and there was a significant relationship between performance at LTP_1_ and work-related strain. Highly fit individuals (>120% P_max-target_) were less prone to overreach the LTP_1_ limit compared to less fit individuals, but standard target exercise performance (100% P_max-target_) seems to be sufficient to stay below this limit on average (Figure 4). Contrary to traditional sports and exercise (except ultra-distance events), heavy occupational work intensity is limited by LTP_1_ performance.

The minimum power output at LTP_1_ for female and male workers in the incremental cycle ergometer test was found to be 60 W (f) and 100 W (m) to stay below the individual LTP_1_ during heavy occupational work with a 10% safety tolerance. This power output at LTP_1_ represents 115 ± 19.6% in the female and 106 ± 23.7% in the male subjects of standard target performance to safely perform heavy occupational work.

## 4. Discussion

Eighty-one subjects presented a normal standard exercise performance (90–110% P_max-target_), 90 subjects were above (>110% P_max-target_) and 69 subjects were below the target (<90% P_max-target_) at the incremental cycle ergometer exercise test. Female workers reached 113.9 ± 19.4% of the target power output, which was significantly higher compared to the male workers who reached 102.5 ± 23.1%. Female workers presented a lower absolute performance than male (Figure 2) and are suggested to have a higher strain during work (Figure 3). This could impose some training effects due to the higher demands for the female subjects during work [34], although no adaptive processes due to heavy work in male workers were shown [35]. Obviously, female workers need to be more fit in relative terms than their male counterparts. Overall, the results of the incremental cycle ergometer exercise tests are within normal limits and comparable to other studies [5].

### 4.1. Measurements during Work

Average HR during the investigated work activities were clearly below the accepted limit of 110 b·min^−1^ for heavy work [10,36]. Studies in steel-workers showed significantly higher HR values with an average HR_8h_ of 108 b·min^−1^ [37], similar to steel-foundry work with HR-values from 112 to 135 b·min^−1^ [11]. These higher HR values in the steel industry are suggested to be caused by higher thermal loads, similar to those observed in firefighters [12]. As no occupations with imposed heat have been included in the study, these comparisons have to be taken with care. Comparable to our study, Anjos et al. [38] and Preisser et al. [13], prescribed a mean HR of 104.0 ± 11.7 and 100.2 ± 11.9 b·min^−1^ for garbage collection work similar to the study by Wakui [39], showing a HR of 98.0 ± 14.0 b·min^−1^ in nurses at day-work. As it was not possible to obtain 8-h oxygen-uptake measures we measured only 20 min of representative work and compared the mean HR during this phase (HR_20_) to the overall 8-h HR (HR_8h_). HR_20_ was significantly related to HR_8h_ (m:r = 0.76, f:r = 0.55, *p* < 0.05) and was therefore regarded as representative for the whole shift as shown earlier [5]. For both female (1.1 b·min^−1^) and male (3.3 b·min^−1^) workers, mean HR_20_ was slightly, but not significantly higher compared to HR_8h_. The slightly higher HR_20_ values can be explained by the fact that the subjects worked continuously during the 20 min measurement period without any interruptions. During eight-hours of working however, there are work-related interruptions and breaks lowering overall mean HR_8h_. Similar result have been shown already by Wultsch et al. [5], confirming our approach. Additionally, the impact of the devices needs to be considered. Carrying a respiratory mask for gas-exchange measurements is an unfamiliar task which may lead to a non-economic breathing accompanied by an increased ventilation, carbon dioxide exhalation and HR [8]. Differences in power output at the LTP_1_ were significantly related to strain levels during the work. As the different strain can also be caused by different workloads chosen freely by the workers, it was an unexpectedly strict relationship indicating that each individual worker was able and allowed to set his workload with respect to their performance capability.

### 4.2. Oxygen Uptake

In the 20 min of VO_2_ measurement during a representative phase of work, female subjects reached a significantly lower mean VO_2_ (0.72 ± 0.25 L·min^−1^) compared to the male workers (1.0 ± 0.3 L·min^−1^). These results are commonly accepted and agree to prescribed limits for heavy occupational work from the literature prescribed as 0.7 L·min^−1^ for female and 1.0 L·min^−1^ for male workers (10,40). Similar results have been shown for male construction workers with 0.92 ± 0.19 L·min^−1^ (14) and for workers of different professions with 0.73 ± 0.29 L·min^−1^ for female and 1.13 ± 0.35 L·min^−1^ for male workers [5]. Callea et al. [37], investigated workers during the apple harvest and found VO_2_ values of 1.37 ± 0.46 L·min^−1^ for male and 1.12 ± 0.46 L·min^−1^ for female harvesters. The results for the male subjects are similar to the present study, but higher for the female subjects. It may be explained by the fact that this work is temporary, and it is suggested that the female workers were somehow forced to keep up with the male workers. Comparing our results to the usual 33% VO_2 max_ limit for heavy work, measures for female (37.8% VO_2 max_) and male (35.3% VO_2 max_) subjects were above the accepted limits [7,40]. As the difference between female and male subjects was not statistically significant, we suggest that workers individually adapt the workload to their individual performance abilities as long as the workload can be chosen freely. This 33% VO_2 max_ limit has been critically discussed and we may suggest this limit to be too low [5], as workers are able to perform their work at higher intensities as shown in our study. Additionally, since this arbitrary value does not represent a physiologically justifiable limit compared to individual thresholds such as the LTP_1_ and VT_1_ [13]. This argument is strengthened by studies in fisherman showing comparable relative VO_2 max_ values to our study at 34–39% [41] and 39% [42], as well as from a study in garbage workers showing a range of 35–69% [13]. Despite the significantly better fitness status of male workers compared to the female, females worked at similar relative values of VO_2 max_. It is therefore obvious that females need a significantly higher fitness level for a similar workload in order to be able to tolerate the same heavy workload [43,44].

### 4.3. Energy Expenditure

Calculated WEE from the 20 min VO_2_ measures of a representative working activity gave 1184 ± 518 kcal (18.1 ± 6.9 MET) for the female and 1660 ± 677 kcal (20.5 ± 8.9 MET) for the male workers, which was significantly different. Similar results have been presented earlier by Wultsch et al. [5], showing results of various occupational groups of heavy work. Comparable to our results WEE was calculated between 1060 kcal and 1790 kcal for an eight-hour shift of 60 steel founders. Although highly intense short stress peaks were prescribed by these authors in this group, the workers were not above the limit for heavy work of 2000 kcal for a working time of eight hours [11]. Significantly higher values were obtained for WEE in 83 male garbage workers summing up to 2240.3 ± 1001.5 kcal for a complete work shift [38]. However, analyzing only WEE during the effective working hours reduced WEE to 1608.3 ± 738.5 kcal (for 293 min) comparable to the present study. Critically we need to mention that WEE in garbage disposal has always to be considered in view of the size of the garbage containers, which makes results from small single studies hard to compare [43]. Dependent on the size of the waste containers, WEE values were shown to be between 2304 kcal and 2784 kcal, which is clearly above the results of the present study, but the lower value is comparable to the study by Anjos et al. [38]. Again, this is an indicator that the limits for heavy occupational work may be accepted higher at least for the short term; however, long-term exposure to such heavy loads may increase the risk for early retirement. To conclude the literature Figure 5 sums up the most common limits accepted to prescribe the limits for heavy work. With exception of one limit for young workers all accepted limits for HR and VO_2_, are clearly below or close to the LTP_1_. Only one absolute value for the HR for young workers is above the LTP_1_ [7]. Correcting this limit for age, again yields a limit below LTP_1_.

### 4.4. Further Calculations and Connections

As the average strain during work in our study indicated by HR and VO_2_ measures was clearly below LTP_1_ and VT_1_ from the incremental cycle ergometer exercise tests one may suggest that this standard exercise test may serve as a measure for heavy occupational work [17]. This is supported by Preisser et al. [13], where absolute and relative values of HR and oxygen consumption of garbage workers measured during work were found below VT_1_. Although most subjects was well below this LTP_1_ limit during their eight-hour working time, additional studies are needed with respect to long-term exposure to such workloads. Interesting to note is the fact that in subjects with the highest fitness levels (>125% P_max-target_), none of the female and male workers were above the LTP_1_ limit which allows to argue a specific exercise performance to be necessary for heavy occupational work. A strong relationship between the fitness level and the strain during work was detected in our large-scale study, which was not expected, but is comparable to other studies [6,45]. As working people are exposed to different workloads and workers can choose workloads freely most of the time this relationship was high but indicated that workers are able to adapt their workload to their individual limits. Thus, apparently less trained persons work slower to stay below the limit of LTP_1_, as confirmed by the work of Astrand [7]. Our results allowed defining a minimum LTP_1_ performance of 100 W (m) and 60 W (f) in an incremental cycle ergometer exercise test. With the help of such a defined minimum power output at the LTP_1_, the risk of acute overload during work can be reduced. However, the long-term risk for cardiovascular disease, overall mortality and early and disability pension needs to be investigated in much more detail [1,3,6].

## 5. Conclusions

Our study shows that the individual exercise performance markers LTP_1_/VT_1_ are valid to prescribe the limits for heavy occupational work. These markers relate workload to the individual strain which may be the main variable to explain acute and chronic overload in heavy workers. Additionally, this threshold allows to prescribe a minimal performance level enabling workers to sustain long-term heavy occupational work.

## Figures and Tables

**Figure 1 jfmk-05-00066-f001:**
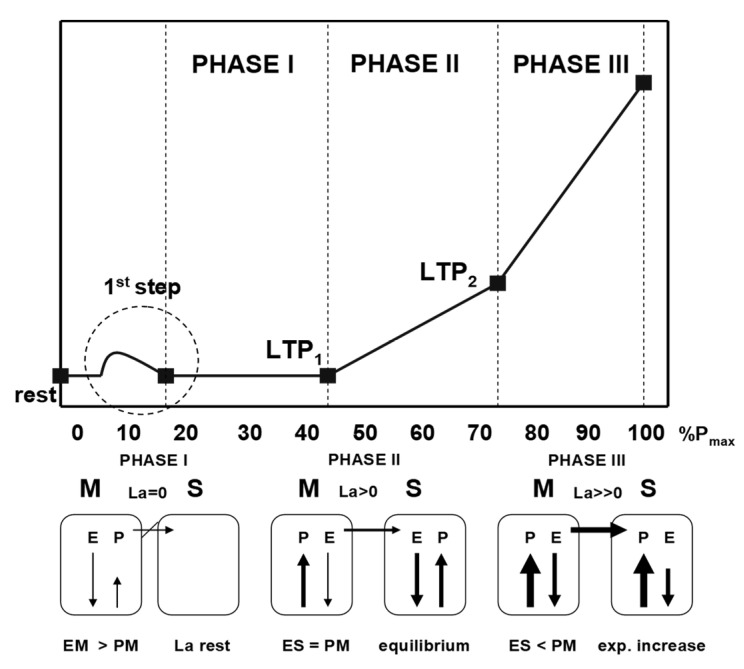
Time course of blood lactate concentration (La) during an incremental cycle ergometer test and the lactate shuttle mechanisms (modified from Tschakert and Hofmann [20]). M—muscle; S—system; P—lactate production; E—lactate elimination; EM—muscular elimination; PM—muscular production; ES—systemic elimination.

**Figure 2 jfmk-05-00066-f002:**
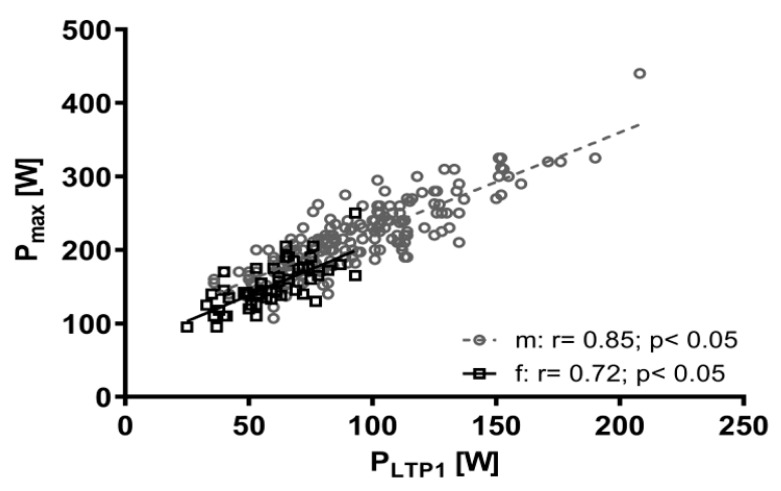
Relationship between power output at the first lactate threshold (P_LTP1_) and maximum power output (P_max_) for male (m) and female (f) workers.

**Figure 3 jfmk-05-00066-f003:**
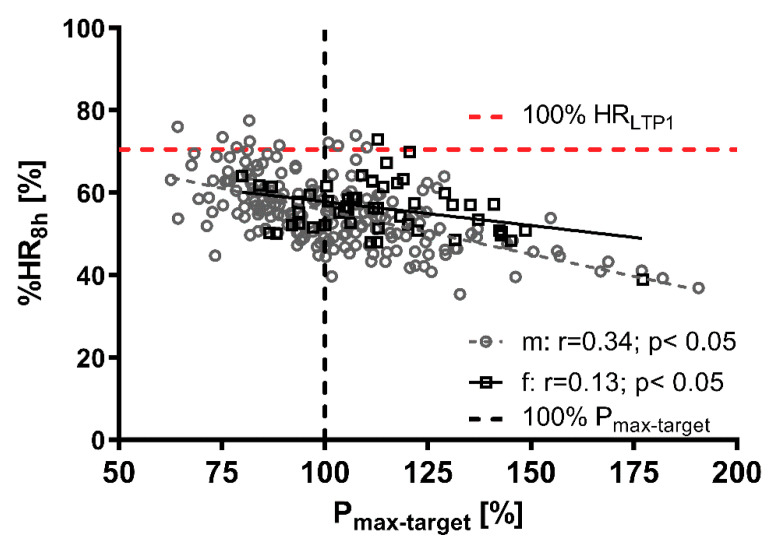
Correlation between the maximum power output (P_max_) related to the target standard P_max_ (P_max-target_) and percentage of maximum heart rate (% HR_8h_) during 8 h occupational work for male (m) and female (f) workers. Dotted lines represent age-predicted maximum target cycle ergometer power output (black) and heart rate at the first lactate turn point (LTP_1_, red).

**Figure 4 jfmk-05-00066-f004:**
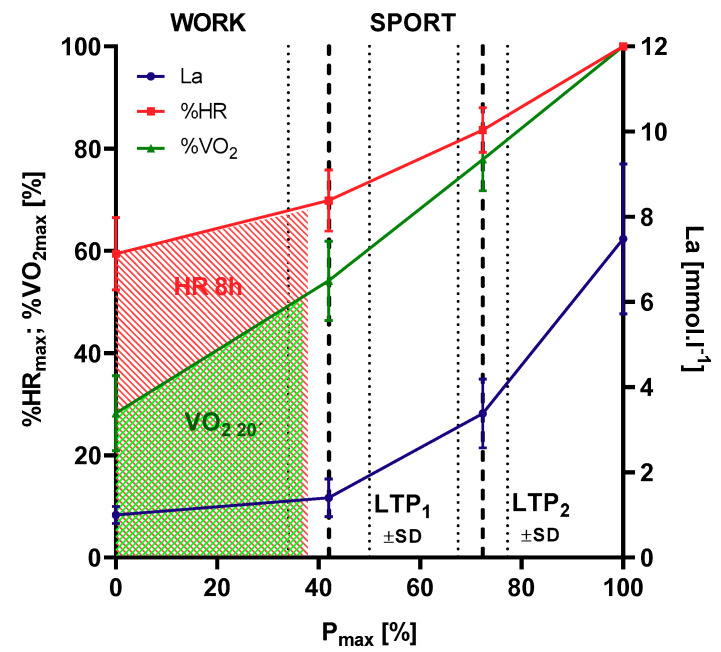
Eight-hour average heart rate (HR, shaded red) and 20-min representative oxygen uptake (VO_2_, shaded green) during 8 h of occupational work related to HR and VO_2_ from incremental cycle ergometer exercise. Phases of metabolism are divided by lactate turn points (LTP_1_, LTP_2_) according to Hofmann et al. [29]. Workloads below LTP_1_ represent occupational work, intensities above LTP_1_ represent sport activities (N = 240).

**Figure 5 jfmk-05-00066-f005:**
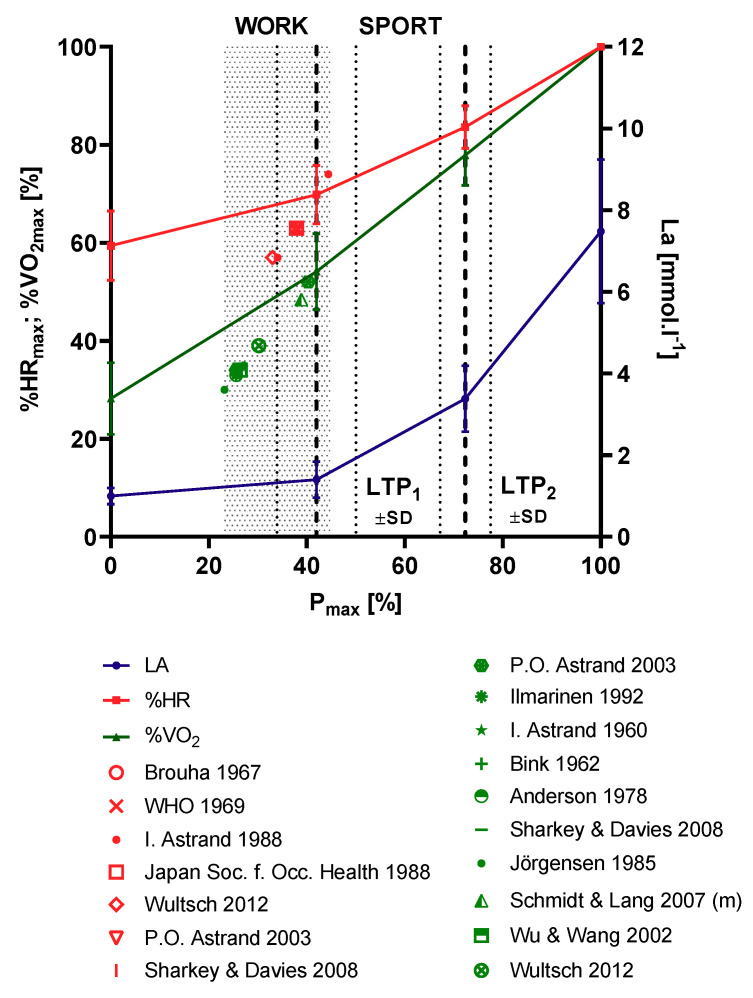
Published limits for heart rate (HR) as a percentage of the maximum HR (% HR_max_, red) and oxygen uptake as a percentage of the maximum oxygen consumption (% VO_2 max_, green) for heavy occupational work related to cycle ergometer exercise reference data. Lactate thresholds (LTP_1_, LTP_2_) were determined according to Hofmann et al. [29], from original data [5]. Workloads below LTP_1_ represent occupational work, strain levels above LTP_1_ are defined as sport activities.

**Table 1 jfmk-05-00066-t001:** Anthropometric data of subjects by sex.

Variables	Female Workers	Male Workers	*p*-Value	Mean Diff. (95% CI)	g
*N*	52	188	–	–	
Age (yrs)	34.5 ± 10.6	34.9 ± 10	0.3	−2.67 (−7.79 to 2.45)	0.04
Height (cm)	165 ± 6	178.3 ± 6.4	<0.0001	0.12 (0.1 to 0.15)	2.11
Weight (kg)	65.3 ± 13	82.1 ± 10	<0.0001	−14.09 (−20.18 to −8.01)	1.57
BMI (kg/m^2^)	23.9 ± 4.3	25.8 ± 4.2	0.12	1.32 (−0.38 to 3.02)	0.45

BMI—body mass index; g—effect size (Hedges’ g); results are shown as *M* ± *SD*; significantly different *p* < 0.05.

**Table 2 jfmk-05-00066-t002:** Maximal and percentages of standard target values of female and male incremental cycle ergometer exercise test results.

Variables	Female Workers	Male Workers	*p*-Value	Mean Diff. (95% CI)	g
P_max_ (W)	150.0 ± 30.8	217.6 ± 48.2	<0.0001	67.65 (56.69 to 78.60)	1.50
% P_max-target_ (%)	113.9 ± 19.4	102.5 ± 23.1	0.0006	−11.33 (−17.63 to −5.02)	−0.51
VO_2 max_ (L·min^−1^)	1.94 ± 0.44	2.90 ± 0.60	<0.0001	0.96 (0.82 to 1.11)	1.69
VO_2 max rel._ (mL·kg^−1^)	30.3 ± 7	36.0 ± 8.2	<0.0001	5.72 (3.46 to 7.97)	0.72
% VO_2 max-target_ (%)	121.4 ± 17.5	114.2 ± 11.7	0.01	−7.18 (−12.33 to −2.04)	−0.55
HR_max_ (b·min^−1^)	177.6 ± 16.1	176.4 ± 14.2	0.63	−1.18 (−9.09 to 3.74)	−0.08
% HR_max-target_ (%)	93.6 ± 8.8	94.8 ± 7.2	0.38	1.17 (−1.46 to 3.81)	0.16
La_max_ (mmol·L^−1^)	6.99 ± 1.47	7.62 ± 1.81	0.01	0.63 (0.14 to 1.11)	0.36
RER_max_	1.20 ± 0.11	1.21 ± 0.10	0.65	0.01 (−0.03 to 0.04)	0.1

P_max_—maximum power output; % P_max-target_—P_max_ as a percentage of target standard power output (P_max-target_) [28]; VO_2 max_—maximum oxygen uptake; VO_2 max rel._—maximum oxygen uptake relative to body weight; % VO_2 max-target_—VO_2 max_ as a percentage of target standard VO_2 max_ [28]; HR_max_—maximum heart rate; % HR_max-target_—HR_max_ as a percentage of target standard HR_max_ [28]; La_max_—maximum lactate concentration; RER_max_—maximum respiratory exchange ratio; g—effect size (Hedges’ g); results are shown as *M* ± *SD*; significantly different *p* < 0.05).

**Table 3 jfmk-05-00066-t003:** Power output (P), oxygen uptake (VO_2_), heart rate (HR) and blood lactate concentration (La) at the first lactate (LTP_1_) and ventilatory threshold (VT_1_) with corresponding percentages to the maximal values.

Variables	Female Workers	Male Workers	*p*-Value	Mean Diff. (95% CI)	g
P_LTP1_ (W)	58.4 ± 14.8	94.1 ± 30.5	<0.0001	35.74 (29.58 to 41.91)	1.28
% P_max_ (%)	38.9 ± 7.4	42.8 ± 8.0	0.001	3.9 (1.55 to 6.25)	0.5
VO_2 LTP1_ (L·min^−1^)	1.09 ± 0.22	1.54 ± 0.37	<0.0001	0.45 (0.37 to 0.53)	1.31
% VO_2 max_ (%)	57.0 ± 8.1	53.3 ± 7.4	0.004	−3.71 (−6.21 to −1.21)	−0.49
HR_LTP1_ (b·min^−1^)	127.4 ± 16.7	122.1 ± 12.4	<0.05	−5.39 (−10.36 to −0.42)	−0.39
% HR_max_ (%)	71.7 ± 5.7	69.3 ± 6.0	<0.0001	−15.83 (−16.89 to −14.78)	−0.4
La_LTP1_ (mmol·L^−1^)	1.34 ± 0.45	1.42 ± 0.44	0.26	0.08 (−0.06 to 0.22)	0.18
P_VT1_ (W)	60.8 ±17.3	98.2 ± 31.6	<0.0001	37.45 (30.76 to 44.14)	1.28
% P_max_ (%)	40.5 ± 8.1	44.5 ± 7.9	<0.001	4.50 (1.97 to 7.04)	0.5
VO_2 VT1_ (L·min^−1^)	1.10 ± 0.25	1.58 ± 0.38	<0.0001	0.48 (0.39 to 0.57)	1.35
% VO_2 max_ (%)	57.0 ± 7.9	54.7 ±5.3	0.06	−2.30 (−4.75 to 0.14)	−0.39
HR_VT1_ (b·min^−1^)	127.7 ± 16.3	123.4 ± 11.9	0.08	−4.30 (−9.17 to 0.57)	−0.33
% HR_max_ (%)	71.8 ± 5.4	70.1 ± 5.3	<0.05	−1.74 (−3.42 to −0.06)	−0.32
La_VT1_ (mmol·L^−1^)	1.55 ± 0.50	1.64 ± 0.41	0.25	0.09 (−0.07 to 0.25)	0.21

P_LTP1_/P_VT1_—power output at the first lactate (LTP_1_) or ventilatory threshold (VT_1_); % P_max_—P_LTP1_/P_VT1_ as a percentage of maximum power output; VO_2 LTP1_/VO_2 VT1_—oxygen uptake at LTP_1_/VT_1_; % VO_2 max_—VO_2 LTP1_/VO_2 VT1_ as a percentage of maximum oxygen uptake; HR_LTP1_/HR_VT1_—heart rate at LTP_1_/VT_1_; % HR_max_—HR_LTP1_/HR_VT1_ as a percentage of maximum heart rate; La_LTP1_/La_VT1_—lactate concentration at LTP_1_/VT_1_; g—effect size (Hedges’ g); results are shown as *M* ± *SD*; significantly different *p* < 0.05).

**Table 4 jfmk-05-00066-t004:** Results of the measurements during a work shift with percentages based on the maximum heart rate (HR_max_) and the heart rate at LTP_1_ (HR_LTP1_).

Variables	Female Workers	Male Workers	*p*-Value	Mean Diff. (95% CI)	g
HR_8h_ (b·min^−1^)	99.3 ± 10.0	96.9 ± 14.1	0.17	−2.41(−5.84 to 1.09)	−0.18
% HR_LTP1_ (%)	78.9 ± 10.2	79.9 ± 12.1	0.52	1.07(−2.24 to 4.38)	0.09
% HR_max_ (%)	56.2 ± 6.2	55.2 ± 8.5	0.33	−1.03 (−3.14 to 1.08)	−0.12
HR_20_ (b·min^−1^)	101.0 ± 12.1	100.2 ± 16.0	0.71	−0.75 (−4.82 to 3.32)	−0.05
% HR_LTP1_ (%)	80.1 ± 11.5	82.7 ± 13.8	0.18	2.54 (1.20 to 6.27)	0.20
% HR_max_ (%)	57.2 ± 7.6	57.1 ± 9.5	0.92	−0.12 (−2.63 to 2.39)	−0.01
VO_2 20_ (L·min^−1^)	0.72 ± 0.25	1.00 ± 0.30	<0.0001	0.28(0.20 to 0.36)	1.00
% VO_2 LTP1_ (%)	67.4 ± 23.9	67.9 ± 26.3	0.89	0.54 (−7.10 to 8.17)	0.02
% VO_2 max_ (%)	37.8 ± 12.3	35.3 ± 11.5	0.19	−2.51 (−6.30 to 1.28)	−0.21
EEr_24h_ (kcal)	1366 ± 174	1766 ± 154	<0.0001	400 (347 to 453)	2.52
EE_8h_ (kcal)	1639 ± 537	2249 ± 691	<0.0001	610 (431 to 789)	0.92
MET_8h_ (EE)	25.2 ± 6.9	27.8 ± 8.9	<0.05	2.57 (0.27 to 4.87)	0.31
WEE_8h_ (kcal)	1184 ± 518	1660 ± 677	<0.0001	477 (304 to 650)	0.74
MET_8h_ (WEE)	25.21 ± 6.91	27.78 ± 8.87	<0.05	2.57 (0.27 to 4.86)	0.3
MET_1h_ (WEE)	3.15 ± 0.86	3.47 ± 1.11	<0.05	0.30 (0.01 to 0.59)	0.3

HR_8h_—heart rate during the eight hours working; HR_20_—heart rate during the 20 min measurement of oxygen uptake; % HR_LTP1_—HR_8h_/HR_20_ as a percentage of the heart rate at the first lactate threshold (LTP_1_); % HR_max_—HR_8h_/HR_20_ as a percentage of maximum heart rate; VO_2 20_—oxygen uptake during the 20 min measurement; % VO_2 LTP1_—VO_2 20_ as a percentage of the oxygen uptake at LTP_1_; % VO_2 max_—VO_2 20_ as a percentage of maximum oxygen uptake; EE_r24h_—24 h resting energy expenditure; EE_8h_—total energy expenditure during eight hours of work; WEE_8h_—working energy expenditure during the eight hours work with the respective metabolic equivalents (MET_8h_/MET_1h_); g—effect size (Hedges’ g); results are shown as *M* ± *SD*; significantly different *p* < 0.05).

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
