# Peer review of "The First Lactate Threshold Is a Limit for Heavy Occupational Work"

_jfmk, 2020, doi:10.3390/jfmk5030066_

Round 1
Reviewer 1 Report
General comment
Thank you for the current study. It is an intriguing study. However a revised needed in a lot of chapters (Abstract, Introduction, Method) and extensively corrections are appropriate.
ABSTRACT
General comment: I think that you need to revise the chapter of Abstract. It is wordy for the readers. Please provide a new version with sufficient purpose, methodology, results and conclusion.
Specific comments
Specific Comment 1
Line 14: It is wordy please revised it.
Specific Comment 2
Line 14-15: The purpose of the study is wordy for the readers. Please, provide a clear purpose.
Specific Comment 3
Line 17: Insert the word "threshold" after "second lactate".
Specific Comment 4
Line 19: I think that you should change the word "gas" and replace it with oxygen uptake or oxygen consumption.
Specific Comment 5
Lines 16-19: You do not provide sufficient explanation of what you did in method. Please revise it.
INTRODUCTION
General comment: I think that you should revised the introduction and to follow a similar structure with the Discussion.
For example, insert paragraphs which each one to correspond in a measured variable and provde a connection between measured variables.
Please careful in grammar and syntax.
Specific Comments
Specific Comment 1
Lines 88-89: Please provide a clear purpose and insert the hypothesis of your study.
Scpecific Comment 2
Figure 1 should move it in the chapter of Method
MATERIAL & METHODS
General comment: A revised is needed in this chapter.
Specific comment 1
Tbale 1 should move it in the chapter of Methodology after the 1st paragraph and you need to provide further information about the participants (i.e. maximum oxygen uptake).
Specific comment 2
Line 98-99: Please define if the participants complete written inform conset. It is extremely important to define this.
Specific comment 3
Line 102: What is ECG please give a definition. Please provide definitions in the abbreviation that you used in the text when report it for the first time.
Specific comment 4
Line 111: As i referred previous in my specific comment 3, please careful with the abbreviations. What is "(m") and what is "(f)"? Please define this.
Specific comment 5
Do you have measured the effect size? Please give an explanation about this.
RESULTS
General comment: You need minor veision in this chapter.
Specific comments
Specific comment 1
Please insert p values, 95% condifence limits and effect size in all tables for each variable separately.
DISCUSSION
General comment: This is the most well written part of your study.
Please careful abou the grammar and syntax in some sentenses.
REFERENCES
Please check again the reference in accordance the journal instructions.
Author Response
ABSTRACT
General comment: I think that you need to revise the chapter of Abstract. It is wordy for the readers. Please provide a new version with sufficient purpose, methodology, results and conclusion.
Response to the General comment: Thank you for your comments. We revised the abstract in order to improve the structure and readability.
Specific comments
Specific Comment 1
Line 14: It is wordy please revised it.
Response: We thank the reviewer for this suggestion. We adapted the text accordingly.
Specific Comment 2
Line 14-15: The purpose of the study is wordy for the readers. Please, provide a clear purpose.
Response: We revised the purpose of the study to improve clarity.
Specific Comment 3
Line 17: Insert the word "threshold" after "second lactate".
Response: As the term “thresholds” after the word “ventilatory” also refers to “the first and second lactate“ we did not include it twice.
Specific Comment 4
Line 19: I think that you should change the word "gas" and replace it with oxygen uptake or oxygen consumption.
Response: Following your comment, we changed the term to “oxygen uptake”.
Specific Comment 5
Lines 16-19: You do not provide sufficient explanation of what you did in method. Please revise it.
Response: Following your comment, we included some more information about the measurements and we included more detailed information in the methods section of the abstract.
INTRODUCTION
General comment: I think that you should revised the introduction and to follow a similar structure with the Discussion.
For example, insert paragraphs which each one to correspond in a measured variable and provide a connection between measured variables.
Response to General comment: Thank you very much for your constructive comment. We are grateful for your suggestion that was very helpful for improving our manuscript. We hope that you will be satisfied with the revised version in which we substantially revised the introduction.
Please careful in grammar and syntax.
Response: We carefully revised the whole manuscript regarding grammar and syntax where necessary.
Specific Comments
Specific Comment 1
Lines 88-89: Please provide a clear purpose and insert the hypothesis of your study.
Response: Following your comment we revised the purpose and included a hypothesis for our study. Lines: 90-93
Specific Comment 2
Figure 1 should move it in the chapter of Method
Response: Figure 1 describes the general physiology of the lactate shuttle theory and their three distinct phases of energy supply. Our hypothesis is explained by this in the introduction part. Therefore, in our opinion placing Figure 1 in the introduction makes it easier for the reader to follow and therefore we like to keep it at this place.
MATERIAL & METHODS
General comment: A revised is needed in this chapter.
Specific comment 1
Table 1 should move it in the chapter of Methodology after the 1st paragraph and you need to provide further information about the participants (i.e. maximum oxygen uptake).
Response: We agree, we moved table 1 into the method section. As VO2max is a result we did not include this information in the method section. This information about participants performance (such as VO2max, HRmax) are presented in Table 2.
Specific comment 2
Line 98-99: Please define if the participants complete written inform consent. It is extremely important to define this.
Response: Thank you for pointing out that this essential part was missing. Before any study procedures were undertaken, the participants completed informed consent and were familiarized with the testing protocol. Following your comment, we included this in the manuscript. (Lines: 105-106)
Specific comment 3
Line 102: What is ECG please give a definition. Please provide definitions in the abbreviation that you used in the text when report it for the first time.
Response: We included the abbreviation for ECG in the text, thank you. (Line: 109)
Specific comment 4
Line 111: As I referred previous in my specific comment 3, please careful with the abbreviations. What is "(m") and what is "(f)"? Please define this.
Response: Thanks’ for this comment, we included the abbreviation earlier in the text. Line: 95
Specific comment 5
Do you have measured the effect size? Please give an explanation about this.
Response: Following your comment regarding the effect size in the result section, we additionally calculated the effect size for groups with different sample size by “Hedges g”. We included this information in the tables as well as statistical analysis.
RESULTS
General comment: You need minor revision in this chapter.
Specific comments
Specific comment 1
Please insert p values, 95% confidence limits and effect size in all tables for each variable separately.
Response: We included the missing information in all tables.
DISCUSSION
General comment: This is the most well written part of your study.
Please careful about the grammar and syntax in some sentences.
REFERENCES
Please check again the reference in accordance the journal instructions.
Response: We revised the references according to the journal’s guidelines.
Reviewer 2 Report
I consider this paper to be a good job to be published in this journal.
Author Response
Thank you very much for reviewing our manuscript!
Reviewer 3 Report
Abstract
Line 14. Please consider rephrasing this statement to have a better link with the aim of the study.
Line 22. Please consider retain only significant findings in the abstract.
Line 26. Make sure conclusions are supported by the results listed in the abstract.
Introduction
The main issue with this section is that it is somehow unbalanced, with the second half (from line 60 on) better developed than the first one. The authors should keep in mind that this section is a development of the hypothesis leading to the aim of the study. At present this should be improved.
Lines 40-44. After the abstract, this is the first impact the reader has with the manuscript. Please try to be as clear as possible, probably using more concise sentences in a consequential order.
Lines 44-50. This part seems to be sort of fragmented. Try to be as clear and smooth as possible in reporting findings from previous literature.
Line 50. Not clear the meaning of “For this reason, physiological limits for heavy work have been set up.”
Line 56. Make sure you spell out abbreviations first before using them. After that you can always use the abbreviations. This should be double checked throughout the manuscript.
Lines 55-59. Please double check this part to improve clarity. At present it is hard to follow and difficult to understand.
Lines 84-86. It seems that something is missing from this sentence.
Line 87. “This knowledge…” Not clear how “This” is related to the previous sentence.
Lines 119-120. Why “anaerobic threshold” is in brackets?
Methods
Line 133. How was the representative work phase established? Given the different nature of the workers this should be clarified.
Results
Line 167. Maybe a reference indicating where the standard results come from could be of help.
Make sure you don’t replicate results in text and tables. Also, Table should stand on their own, so a footnote with abbreviation should be included.
Figure 2. As most of the female results are hidden by the males, will it be better to have them separate? Maybe as Figure 2a and 2b? The same applies for Figure 3.
Is there any information available about the training regimen of the subjects?
References are not in the proper journal format.
Author Response
Abstract
Comment: Line 14. Please consider rephrasing this statement to have a better link with the aim of the study.
Response: According to reviewer 1 and your comment, we revised the abstract accordingly.
Comment:Line 22. Please consider retain only significant findings in the abstract.
Response: As it is important to show that work strain is not different between male and female workers, we like to keep this non-significant difference in the abstract.
Comment:Line 26. Make sure conclusions are supported by the results listed in the abstract.
Response: As more or less all workers stayed below LTP1 intensity during their work, we suggest that our conclusion is supported by our presented findings.
Introduction
Comment: The main issue with this section is that it is somehow unbalanced, with the second half (from line 60 on) better developed than the first one. The authors should keep in mind that this section is a development of the hypothesis leading to the aim of the study. At present this should be improved.
Response: According to reviewer 1 and your comment, we substantially reworded the introduction section.
Comment: Lines 40-44. After the abstract, this is the first impact the reader has with the manuscript. Please try to be as clear as possible, probably using more concise sentences in a consequential order.
Response: Following your comment, we revised this paragraph. Line 40-44
Comment: Lines 44-50. This part seems to be sort of fragmented. Try to be as clear and smooth as possible in reporting findings from previous literature.
Response: Following your comment, we reworded the paragraph to improve clarity. Line 44-55
Comment: Line 50. Not clear the meaning of “For this reason, physiological limits for heavy work have been set up.”
Response: We removed the term: “for this reason” and specify the sentence. Line 48
Comment: Line 56. Make sure you spell out abbreviations first before using them. After that you can always use the abbreviations. This should be double checked throughout the manuscript.
Response: Thanks for this comment, we controlled all abbreviations throughout the text
Comment: Lines 55-59. Please double check this part to improve clarity. At present it is hard to follow and difficult to understand.
Response: Following your comment, we revised this paragraph to improve clarity. Line: 60-63
Comment: Lines 84-86. It seems that something is missing from this sentence.
Response: Thank you for this comment, we revised this sentence. Line: 87-89
Comment: Line 87. “This knowledge…” Not clear how “This” is related to the previous sentence.
Response: We removed the term: “This knowledge” and specify the sentence. Line 89
Comment: Lines 119-120. Why “anaerobic threshold” is in brackets?
Response: As this term is differentially used in the USA and Europe, with completely different meanings (USA: first increase in blood lactate concentration; Europe: second threshold according to maximal lactate steady state) we try to avoid this term as far as possible. To highlight this situation, we used the term in brackets here.
Methods
Comment: Line 133. How was the representative work phase established? Given the different nature of the workers this should be clarified.
Response: Due to the repetitive character of the work, it was easy to define a representative work phase in our subjects. We controlled this 20 minute phase by 20 minute mean heart rate to 8 hours mean heart rate to be sure to be representative.
Results
Comment: Line 167. Maybe a reference indicating where the standard results come from could be of help.
Response: The reference was already in the text, but we included this reference in the method section also. (Line: 139)
Comment: Make sure you don’t replicate results in text and tables. Also, Table should stand on their own, so a footnote with abbreviation should be included.
Response: Thanks for your comment, footnotes have been included accordingly.
Comment: Figure 2. As most of the female results are hidden by the males, will it be better to have them separate? Maybe as Figure 2a and 2b? The same applies for Figure 3.
Response: As it is important to see possible differences between male and female workers we kept data within one figure, but changed the symbols for better visibility.
Comment: Is there any information available about the training regimen of the subjects?
Response: We did not obtain any information on physical activity or training, however, performance of subjects (Pmax, VO2max) indicate a different training status/ fitness level.
Comment: References are not in the proper journal format.
Response: We revised the references according to the journal’s guidelines.
Round 2
Reviewer 1 Report
I want to thank the authors for their grateful work to reconstruct the current paper. All my concerns have been solved.
It is an intriguing work.
Author Response
We thank the reviewer for the valuable review-process
Reviewer 3 Report
To be consistent, I suggest the authors to use the same symbols for male and female in Figure 2 and 3. In particular, as it seems to be clearer, I suggest to use the same format as Figure 3 (grey circles for males).
Author Response
We thank the reviewer for the suggestion and the valuable and helpful review process. The figures have been adapted accordingly (also the regression lines).